# Controllable Heterogeneous Model Aggregation for Personalized Federated Learning

**Jiaqi Wang**[1]  **Qi Li**[2]  **Lingjuan Lyu**[3]  **Fenglong Ma**[1]*

[1]The Pennsylvania State University   [2]Iowa State University   [3]Sony AI

`{jqwang,fenglong}@psu.edu, qli@iastate.edu, lingjuan.lv@sony.com`

## Abstract

Federated learning, a pioneering paradigm, enables collaborative model training without exposing users' data to central servers. Most existing federated learning systems necessitate uniform model structures across all clients, restricting their practicality. Several methods have emerged to aggregate diverse client models; however, they either lack the ability of personalization, raise privacy and security concerns, need prior knowledge, or ignore the capability and functionality of personalized models. In this paper, we present an innovative approach, named `pFedClub`, which addresses these challenges. `pFedClub` introduces personalized federated learning through the substitution of controllable neural network blocks/layers. Initially, `pFedClub` dissects heterogeneous client models into blocks and organizes them into functional groups on the server. Utilizing the designed `CMSR` (Controllable Model Searching and Reproduction) algorithm, `pFedClub` generates a range of personalized candidate models for each client. A model-matching technique is then applied to select the optimal personalized model, serving as a teacher model to guide each client's training process. We conducted extensive experiments across three datasets, examining both IID and non-IID settings. The results demonstrate that `pFedClub` outperforms baseline approaches, achieving state-of-the-art performance. Moreover, our model insight analysis reveals that `pFedClub` generates personalized models of reasonable size in a controllable manner, significantly reducing computational costs[2].

## 1 Introduction

Federated learning (FL) [1, 2, 3, 4, 5, 6, 7, 8, 9, 10, 11, 12, 13, 14, 15] is a prevalent method to train machine learning models collaboratively without centralizing clients' data on a cloud server. However, many current FL training frameworks demand uniformity in deep neural network structures among client models, a requirement often too stringent for practical, real-world applications. An alternative approach involves equipping clients with heterogeneous models, introducing a new challenge: *how to aggregate these diverse models within the federated learning framework effectively*.

Recently, various approaches have emerged to address the challenge of aggregating heterogeneous models, particularly in the context of personalized FL. Some methods leverage *additional information*, such as class information [16], logits [17, 18], and label-wise representations [19], as intermediaries for exchanging information between clients and the server. While seemingly straightforward, this approach raises significant privacy concerns, especially regarding the potential exposure of sensitive client data. To mitigate these privacy concerns, techniques such as *distillation* [20, 21, 22] and *model reassembly* [23] have been introduced in heterogeneous FL, wherein only model parameters are exchanged, akin to traditional FL approaches [1]. Despite demonstrating effectiveness in aggregating

---

*Corresponding author.

[2]The source code can be found at `https://github.com/JackqqWang/24club`.

heterogeneous models, both distillation and model reassembly techniques in FL suffer from a common drawback – **lack of control over personalized model generation**.

Distillation-based approaches inherently necessitate the establishment of a unified model as the global model, informed by prior insights [21]. This global model is then distributed to clients to guide their training efforts. However, a smaller consensus global model may struggle to extract heterogeneous knowledge from clients, often resulting in a larger size. This poses challenges for smaller clients with limited computational resources to run the shared large global model effectively. Similarly, model reassembly-based approaches also encounter a related issue. While they generate personalized models for each client, these personalized models may be significantly larger than the clients' capacity, posing challenges for implementation.

To further investigate these limitations, we conducted an experiment using the state-of-the-art model reassembly approach, pFedHR [23], on the SVHN dataset, where each client operated a distinct model. (*Additional details and further discussions can be found in Section 4.4.*) We evaluated the parameter size of the original models and the average number of parameters in the personalized models received across communication rounds. Subsequently, we illustrated the parameter size differences between these two types of models in Figure 1. The entire bar represents the average parameter size of the generated personalized model, with the original model size depicted in blue and the increased size shown in red. Our preliminary findings indicate that the personalized models produced by pFedHR are notably larger than the original local models.

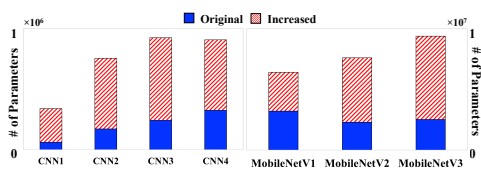

Figure 1: Lack of controllability demonstration using pFedHR [23] on the SVHN dataset with seven different client models by comparing the original model size (blue bars) to the generated personalized client model size (entire bars).

To address this issue, in this paper, we introduce a novel approach for heterogeneous model aggregation, named pFedClub, aiming to achieve **p**ersonalized **Fed**erated learning through **C**ontrollable neural network b**l**ock s**ub**stitution, as depicted in Figure 2. pFedClub receives heterogeneous models uploaded from clients on the server. pFedClub first decomposes the heterogeneous client models into different blocks and subsequently clusters these blocks based on their functionalities (refer to Section 3.2.1). pFedClub explores a novel neural network block substitution technique to achieve this objective, as detailed in Section 3.2.2. Specifically, pFedClub aims to substitute the $r$-th block $\mathbf{B}_{m,r}^t$ within the $m$-th client model $\mathbf{w}_m^t$ with a block selected from the same group as $\mathbf{B}_{m,r}^t$ during communication round $t$. This approach ensures both the functionality of personalized models and their similarity to the original models. To enhance the diversity of the generated models, we permit arbitrary substitutions from the group for the first block (**Step 1**). For subsequent blocks, we introduce *an order-constrained block search strategy* (**Step 2**), ensuring the quality of the generated models. If the order constraint halts the substitution prematurely, the remaining blocks are directly added to the substituted ones (**Step 3**). This completion strategy not only reduces the number of newly added parameters in the stitching blocks but also ensures similar functionality. As pFedClub may generate multiple personalized candidate models for each client, we employ the *similarity-based model-matching* technique to select the personalized model, as discussed in Section 3.2.3. The selected personalized model is then distributed to the respective client as a teacher model, guiding the client model training process through knowledge distillation.

It is essential to emphasize that the proposed pFedClub framework is designed to be both general and flexible, allowing for the incorporation of strict controllable constraints during the personalized candidate generation process. For instance, constraints such as the model size of the generated candidates can be easily integrated into the framework. Furthermore, introducing controllability within pFedClub enables the mitigation of computational and communication costs compared to current model reassembly-based approaches. By incorporating controllable constraints, pFedClub offers greater adaptability and efficiency in handling personalized model generation. Experimental results demonstrate that pFedClub achieves state-of-the-art performance on three benchmark datasets under both IID and non-IID settings, demonstrating the effectiveness of the proposed aggregation strategy.

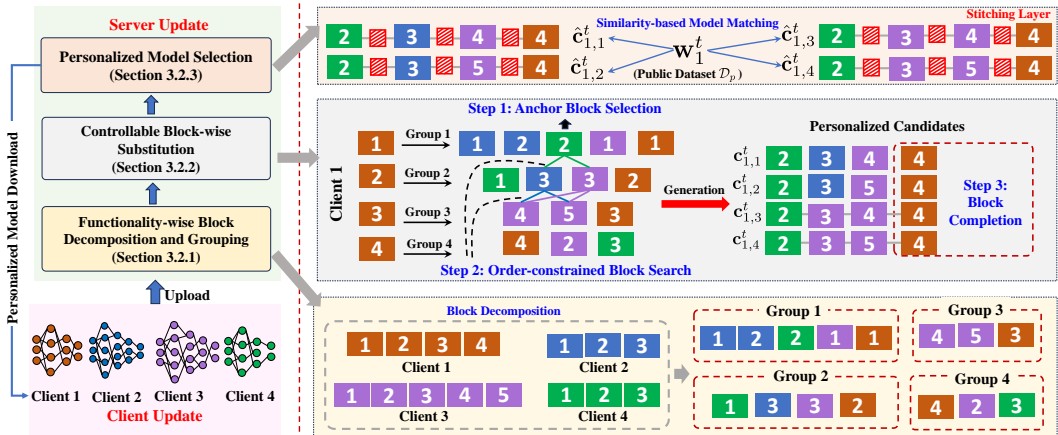

Figure 2: Overview of the proposed `pFedClub`. We take four clients with heterogeneous models as an example. The numbers denote the blocks' indexes. The number of functional groups is 4. We take Client 1 as an example to demonstrate how `pFedClub` works to generate personalized candidate models. Note that the arbitrary substitution for Block 1 of Client 1 is the second block from Client 4.

## 2  Related Work

**Model Heterogeneity in Federated Learning.** Heterogenous model cooperation is a challenging task in FL. Researchers have explored submodel training techniques [24, 25], focusing on training a shared large global model by sending masked heterogeneous models to appropriate clients. However, these approaches often fail to provide personalized models for individual clients. Furthermore, they are considerably constrained by the limitations in freedom of model selection. In addition, FedDF [20] and FedKEMF [21] conduct ensemble distillation, but the settings of FedDF are different from ours and not for model personalization. FedKEMF utilizes mutual knowledge distillation with the requirement of predefined model structures. Besides, HeteroFL [24] and FlexiFed [25] are restricted to the requirements of the client model structures. Other related research work needs extra information to be exchanged between the server and clients, e.g., logits in FCCL [17], class scores in FedMD [16], and label-wise representation in FedGH [19], which raises the concerns of privacy [26]. The most recent work pFedHR [23] provides a layer-wise model reassembly approach to solve the challenge of model heterogeneity in federated learning. However, it has several limitations, as we discuss in Section 1.

**Personalized Federated Learning.** Instead of maintaining one global model, personalized FL cares more about each local model's performance, which is more sufficient and practical. In [27], the authors add a proximal term to the local optimization loss function to bound the difference between the local and global model updates. The aggregated global model is treated as the initial shared model from the meta-learning perspective in [28]. In [29], the authors design a new regularized client loss to optimize the local model to achieve personalization. However, the discussed personalized federated learning work assumes that the clients have to share identical model structures.

## 3  Methodology

### 3.1  Overview

Let $\mathcal{D}_n = \{(\mathbf{x}_i, \mathbf{y}_i)\}_{i=1}^{|\mathcal{D}_n|}$ denote the training data stored in the $n$-th local client $L_n$, where $\mathbf{x}_i$ denotes the data, $\mathbf{y}_i$ is the ground truth, and $|\mathcal{D}_n|$ is the number of training data. Each client employs a deep neural network-based model $\mathbf{w}_n$ to train on its training data. Note that the client models $\{\mathbf{w}_1, \cdots, \mathbf{w}_N\}$ do not share identical network structures, where $N$ is the total number of clients. The overview of the proposed `pFedClub` is depicted in Figure 2, containing server update and client update. During each communication round $t$, the randomly chosen $M$ ($M \ll N$) client models $\{\mathbf{w}_1^t, \cdots, \mathbf{w}_M^t\}$ will be uploaded to the server to generate their personalized models $\{\mathbf{w}*_1^t, \cdots, \mathbf{w}*_M^t\}$. These personalized models will be sent to the corresponding clients as teachers

to guide the update of client models $\{\mathbf{w}_1^{t+1}, \cdots, \mathbf{w}_M^{t+1}\}$ in the next communication round. Next, we provide the details of the model design.

## 3.2 Server Update

The model heterogeneity of the uploaded $M$ client models $\{\mathbf{w}_1^t, \cdots, \mathbf{w}_M^t\}$ at the $t$-th communication round makes it challenging to generate personalized client teachers $\{\mathbf{w}*_1^t, \cdots, \mathbf{w}*_M^t\}$. To tackle this challenge, a new controllable block-wise substitution-based personalized model aggregation approach is proposed, which not only maintains the functionality of each block in the originally uploaded client models but also injects new knowledge provided by other models to further achieve model personalization. To this end, we first divide each client model $\mathbf{w}_m^t$ into blocks and then group blocks into different functionality clusters.

### 3.2.1 Functionality-wise Block Decomposition and Grouping

**Block Decomposition**. Except for recurrent deep neural networks such as the long-short term memory network [30], most of the remaining ones, such as the family of convolutional neural networks (CNN), can be treated as block-stacked neural networks[3]. Let $R_m$ denote the number of blocks in client model $\mathbf{w}_m^t$. We then decompose $\mathbf{w}_m^t$ into blocks $\{\mathbf{B}_{m,1}^t, \cdots, \mathbf{B}_{m,r}^t, \cdots, \mathbf{B}_{m,R_m}^t\}$. Note that a block is either a convolutional net, a fully connected layer, or a building block associated with a shortcut connection such as residual neural networks [31].

**Block Grouping**. After decomposing each model, we then apply the $K$-means algorithm to group blocks based on their functionality. Specifically, we apply the centered kernel alignment (CKA) technique [32] to calculate the similarity between two blocks as follows:

$$\mathbf{sim}(\mathbf{B}_{m,i}^t, \mathbf{B}_{n,j}^t) = \mathbf{CKA}(\mathbf{x}_{m,i}^t, \mathbf{x}_{n,j}^t) + \mathbf{CKA}(\mathbf{B}_{m,i}^t(\mathbf{x}_{m,i}^t), \mathbf{B}_{n,j}^t(\mathbf{x}_{n,j}^t)), \tag{1}$$

where $\mathbf{x}_{\cdot,i}^t$ denotes the input of the block $\mathbf{B}_{\cdot,i}^t$, and $\mathbf{B}_{\cdot,i}^t(\mathbf{x}_{\cdot,i}^t)$ represents the output of the block $\mathbf{B}_{\cdot,i}^t$ ($\cdot = m$ or $n$). Let $\{\mathcal{G}_1^t, \cdots, \mathcal{G}_K^t\}$ denote the $K$ functionality groups generated by the $K$-means algorithm, where each group $\mathcal{G}_k^t$ contains multiple blocks from different models with similar functions.

### 3.2.2 Controllable Block-wise Substitution

Intuitively, if each pair of corresponding blocks of two models has similar functionality, the whole models should also be similar. Based on this straightforward intuition, we propose to replace each block $\mathbf{B}_{m,r}^t$ with a function-similar one chosen from the group $\mathcal{G}_k^t$, where $\mathbf{B}_{m,r}^t \in \mathcal{G}_k^t$. Let $G_k$ denote the number of function-similar blocks in each group $\mathcal{G}_k^t$. Arbitrarily replacing each block without any constraints will produce a vast number of candidate models, which is approximately equal to $G_k^{R_m}$, where $R_m$ is the number of blocks in model $\mathbf{w}_m^t$. It is time-consuming to update all candidates.

To reduce the size of the candidate model pool, a naive solution is to randomly choose a fixed number of models from the pool first and then use the model similarity score to select the personalized teacher model. Although this approach increases the diversity of the candidate model generation, it is uncontrollable to introduce too much randomness, and the low-quality candidates may reduce the convergence rate of the federated system training. To solve these issues, we design a Controllable Model Searching and Reproduction (CMSR) algorithm (as depicted in Algorithm 1), which is a greedy-based search approach to reproduce a set of personalized candidate models for each $\mathbf{w}_m^t$ by considering both **diversity** and **quality**. In particular, CMSR consists of three key steps: anchor block selection, order-constrained block search, and block completion. Next, we describe the details of each step.

**Step 1: Anchor Block Selection** (Alg. 1 lines 3-7). Assume that the first block $\mathbf{B}_{m,1}^t$ in $\mathbf{w}_m^t$ belongs to the group $\mathcal{G}_k^t$. CMSR then randomly selects one block in $\mathcal{G}_k^t$ as the substitution. The substituted block will be treated as the anchor/starting block of all the candidate models. As shown in Figure 2, $\mathbf{w}_{4,2}^t$ is selected as the substitution of $\mathbf{w}_{1,1}^t$. Note that a naive solution to select the anchor block may use the block with the largest similarity score calculated via Eq. (1) instead of randomly choosing one. However, such a solution significantly reduces the diversity of the generated candidates, further limits the extra knowledge borrowed from other models, and finally hurts the personalization of teacher models. Besides, $\mathbf{B}_{m,1}^t$ has a chance to be selected with a probability $\frac{1}{G_k}$.

---

[3]In this paper, we do not decompose recurrent deep neural networks, which is our future work.

---

**Algorithm 1:** The `CMSR` Algorithm

---

    **input**   :Client model $\mathbf{w}_m^t$ and block clusters $\{\mathcal{G}_1^t, \cdots, \mathcal{G}_K^t\}$
    **output** :Candidate set $\{\mathbf{c}_{m,1}^t, \cdots, \mathbf{c}_{m,S_m}^t\}$

**1**   Initialize candidate blocks $\mathcal{C}_m^t = \{\}$;
**2**   **for** $r \leftarrow 1, \cdots, R_m$ **do**
**3**      // Step 1: Anchor Block Section
**4**      **if** *r = 1* **then**
**5**         Randomly select one block $\mathbf{B}_{p,q}^t$ from the group that contains $\mathbf{B}_{m,1}^t$;
**6**         Add the substituted block to $\mathcal{C}_m^t[1] = [\mathbf{B}_{p,q}^t]$;
**7**         Record the block index $q$;
**8**      // Step 2: Order-constrained Block Search
**9**      **if** *r > 1* **then**
**10**        Initialize block index set $\mathcal{I}_{m,r}^t = []$;
**11**        // Assume that $\mathbf{B}_{m,r}^t \in \mathcal{G}_k^t$
**12**        **for** $\mathbf{B}_{\cdot,u}^t \in \mathcal{G}_k^t$ **do**
**13**           **if** $u > q$ **then**
**14**             Add the block to $\mathcal{C}_m^t[r]$;
**15**             Add the block index to $\mathcal{I}_{m,r}^t$;
**16**        **if** $\mathcal{I}_{m,r}^t = \emptyset$ **then**
**17**          break;
**18**        $q \leftarrow \min(\mathcal{I}_{m,r}^t)$;
**19** // Candidate Generation
**20** Use $\mathcal{C}_m^t$ to generate candidates that satisfy the order condition;
**21** **if** *the number of blocks of the candidate model is smaller than* $R_m$ **then**
**22**      Run Step 3: Block Completion to complete the remaining blocks;

     **return :** $\{\mathbf{c}_{m,1}^t, \cdots, \mathbf{c}_{m,S_m}^t\}$

---

**Step 2: Order-constrained Block Search** (Alg. 1 lines 8-18). After selecting the anchor block in Step 1, `CMSR` then finds the substitutions for the following blocks. The simplest solution is repeating the previous step $R_m - 1$ times to generate a candidate model. As we discussed before, it may generate low-quality candidate models.

To avoid this problem and generate controllable high-quality candidates, we maintain the order of the selected blocks, even from different client models, as a hard constraint [33]. Mathematically, let $\mathbf{B}_{p,q}^t$ denote the substitution of the $r$-th block of $\mathbf{B}_{m,r}^t$, where $p$ is the model index and $q$ is the block index. For any substitution of the $(r+1)$-th block $\mathbf{B}_{\cdot,u}^t$ should satisfy the constraint $q < u$. As shown in Figure 2, the substitutions of $\mathbf{w}_{1,2}^t$ include $\mathbf{w}_{2,3}^t$ and $\mathbf{w}_{3,3}^t$ since the index of the first block's substitution $\mathbf{w}_{4,2}^t$ is 2.

Only using the order constraint is sufficient for the proposed `pFedClub` to generate high-quality and informative candidates. First, maintaining the order of block functions can guarantee that the candidate model has functionality similar to the original model. Second, it also avoids pre-defining the order of operation types, which releases the constraints of personalized teacher model generation and further increases the diversity of candidates. Third, such an approach is capable of generating similar-sized personalized models for clients. It is essential for several applications with limited computational resources, such as smart devices. However, pFedHR cannot control the size of the generated candidates.

*It is worth noting that, aside from the order constraint, `pFedClub` is highly adaptable and can easily incorporate other types of constraints. In Section 4.4, we will delve into the integration of the model size constraint, demonstrating the framework's versatility and ability to accommodate various constraints for personalized model generation.*

**Step 3: Block Completion** (Alg. 1 lines 19-22). Step 2 may stop at the certain block $r < R_m$ due to the block order constraint. To maintain the original functional structure, we will add the remaining blocks of $\mathbf{w}_m^t$, i.e., $\{\mathbf{B}_{m,r+1}^t, \cdots, \mathbf{B}_{m,R_m}^t\}$ to the substitutions. In such a way, `pFedClub` can

generate a set of candidates denoted as $\{\mathbf{c}_{m,1}^t, \cdots, \mathbf{c}_{m,S_m}^t\}$, where $S_m$ is the number of generated candidate models. As shown in Figure 2, pFedClub will stop after substituting the third block $\mathbf{w}_{1,3}^t$ since all the block indexes in Group 4 for substituting $\mathbf{w}_{1,4}^t$ are not greater than 4, which is the minimum feasible block index. Thus, pFedClub will complete the generated candidates by directly using $\mathbf{w}_{1,4}^t$ as the forth block.

### 3.2.3 Personalized Model Selection

The final stage of pFedClub is to automatically select the "best" candidate teacher model from $\{\mathbf{c}_{m,1}^t, \cdots, \mathbf{c}_{m,S_m}^t\}$ for the $m$-th client. However, selecting such a model is non-trivial because $\mathbf{c}_{m,s}^t$ is a reassembled, incomplete model via block substitution, and the dimension sizes of different blocks may not be well-aligned.

**Block Stitching**. We complete each candidate model $\mathbf{c}_{m,s}^t$ using the network stitching technique [34]. We use a nonlinear activation function ReLU$(\cdot)$ on top of a linear layer, i.e., ReLU$(\mathbf{W}^\top \mathbf{X} + \mathbf{b})$ as the dimension mapping function.

Since the parameter values of $\{\mathbf{W}, \mathbf{b}\}$ in the stitching functions are **unknown**, it is essential to learn them with training data. Here, we propose to use a public dataset $\mathcal{D}_p$ to fine-tune the stitched candidate model $\mathbf{c}'^t_{m,s}$. Note that we fix all the parameters in $\mathbf{c}_{m,s}^t$ (denoted as $\boldsymbol{\theta}_{m,s}^*$) and only update $\{\mathbf{W}, \mathbf{b}\}$ in $\mathbf{c}'^t_{m,s}$ using the following loss if the public data are labeled:

$$\mathcal{L}_m = \frac{1}{|\mathcal{D}_p|} \sum_{i=1}^{|\mathcal{D}_p|} \mathrm{CE}(\mathbf{c}'^t_{m,s}(\mathbf{x}_i; \mathbf{W}, \mathbf{b}, \boldsymbol{\theta}_{m,s}^*), \mathbf{y}_i), \tag{2}$$

where $|\mathcal{D}_p|$ denotes the number of data in the public dataset, CE$(\cdot, \cdot)$ means the cross-entropy loss, $\mathbf{c}'^t_{m,s}(\mathbf{x}_i; \mathbf{W}, \mathbf{b}, \boldsymbol{\theta}_{m,s}^*)$ presents the predicted label distribution for the data $\mathbf{x}_i$ by fixing the parameters $\boldsymbol{\theta}_{m,s}^*$, and $\mathbf{y}_i$ is the ground truth vector.

An unlabeled public dataset can be used to fine-tune the parameters $\{\mathbf{W}, \mathbf{b}\}$ using the normalized temperature-scaled cross-entropy loss [35] for a pair of data as follows:

$$\mathcal{L}_m^{i,j} = -\log \frac{\exp(\cos(\mathbf{c}'^t_{m,s}(\mathbf{x}_i), \mathbf{c}'^t_{m,s}(\mathbf{x}_j))/\tau)}{\sum_{k=1}^{2|\mathcal{D}_p|} \mathbb{1}_{[k \neq i]} \exp(\cos(\mathbf{c}'^t_{m,s}(\mathbf{x}_i), \mathbf{c}'^t_{m,s}(\mathbf{x}_k))/\tau)}, \tag{3}$$

where $\cos(\cdot, \cdot)$ is the cosine similarity, and $\tau$ is the hyperparamter. $\mathbf{x}_j$ is the augmentation of $\mathbf{x}_i$. We still fix the parameters $\boldsymbol{\theta}_{m,s}^*$ and learn $\{\mathbf{W}, \mathbf{b}\}$.

It is worth noting that block stitching operation will not significantly increase the number of parameters in the candidate model. Besides, some candidate models may be generated using Step 3: block completion. For those candidates, the number of newly added parameters is much smaller. Moreover, the limited number of parameters is helpful for the new candidate models to maintain more original model information. Finally, it makes model computation efficient and speeds up the model training.

**Model Selection**. Let $\hat{\mathbf{c}}_{m,s}^t$ denote the fine-tuned candidate model via Eq. (2). We then calculate the average cosine similarity scores on logits outputted by the original model $\mathbf{w}_m^t$ and its candidate model $\hat{\mathbf{c}}_{m,s}^t$ as follows:

$$\alpha_{m,s} = \frac{1}{|\mathcal{D}_p|} \sum_{i=1}^{|\mathcal{D}_p|} \cos(\boldsymbol{\beta}_m^t(\mathbf{x}_i), \hat{\boldsymbol{\beta}}_{m,s}^t(\mathbf{x}_i)), \tag{4}$$

where $\boldsymbol{\beta}_m^t(\mathbf{x}_i)$ and $\hat{\boldsymbol{\beta}}_{m,s}^t(\mathbf{x}_i)$ denote the logits of $\mathbf{x}_i$ outputted by the models $\mathbf{w}_m^t$ and $\hat{\mathbf{c}}_{m,s}^t$, respectively. Note that this is a forward propagation and does not need to train the models. Finally, the candidate model with the highest similarity scores in the set $\{\alpha_{m,1}, \cdots, \alpha_{m,S_m}\}$ will be selected as the final personalized teacher model $\mathbf{w}*_m^t$.

### 3.3 Client Update

When the $m$-th client is selected again at the $j$ $(j > t)$ communication round, the personalized model $\mathbf{w}*_m^t$ generated in the recent communication round will be distributed. Since the teacher model $\mathbf{w}*_m^t$

usually has a different network structure from the client model $\mathbf{w}_m^j$, we propose to use knowledge distillation to update the client model following [36] by optimizing the following loss:

$$\mathcal{L}_n^j = \frac{1}{|\mathcal{D}_n|} \sum_{i=1}^{|\mathcal{D}_n|} \left[ \mathrm{CE}(\mathbf{w}_m^j(\mathbf{x}_i), \mathbf{y}_i) + \lambda \mathrm{KL}(\boldsymbol{\beta}_m^j(\mathbf{x}_i), \hat{\boldsymbol{\beta}}_m^t(\mathbf{x}_i)) \right], \tag{5}$$

where $\lambda$ is a hyperparameter and $\mathrm{KL}(\cdot, \cdot)$ is the Kullback–Leibler divergence.

## 4 Experiments

### 4.1 Experimental Setups

**Datasets**. In our experiments, we utilize three commonly used datasets to validate the performance of the proposed `pFedClub`, including MNIST[4], SVHN[5], and CIFAR-10[6]. We randomly divide the datasets into three parts: 72% for training, 20% for testing, and 8% as the public dataset. We test two data distribution settings in federated learning, i.e., IID and non-IID, following existing work [23]. For the **IID** setting, the training and testing data are randomly distributed to $N$ clients. For the **non-IID** setting, each client randomly holds data belonging to two classes.

**Baselines**. The proposed `pFedClub` aims to aggregate **heterogeneous** client models to boost federated learning performance. Based on the condition of public datasets, we consider the following approaches as our baselines: (1) without using public datasets: HeteroFL [24] and FlexiFed [25]; (2) using labeled public data: FedMD [16], FedGH [19], and pFedHR [23]; and (3) using unlabeled public data: FCCL [17], FedKEMF [21], and pFedHR [23]. We also compare the proposed `pFedClub` with the general approaches to learning personalized federated learning models, which share the same structure for all clients. The **homogeneous** baselines include: FedAvg [1], FedProx [27], Per-FedAvg [28], PFedMe [29], PFedBayes [37], and pFedHR [23]. The details of all baselines can be found in Appendix **A**.

**Client Model Deployment**. In our experiments, we employ seven client models with different network structures, including MobileNetV1 [38], MobileNetV2 [39], MobileNetV3 [40], and four manually designed CNN models (denoted as CNN1 to CNN4). Each CNN model contains several convolutional blocks and fully connected blocks. The detailed model structures of the four models can be found in Appendix **B**. We set the number of clients $N = 50$ and the number of active clients $M = 5$ in each communication round. We propose three plans to distribute client models to validate the performance of the proposed `pFedClub` in different scenarios. First, we use all seven types of models and randomly send each model to a client (**Model Zoo I**). Specifically, CNN1 is randomly assigned to 8 clients, and each of the remaining six models is randomly assigned to 7 clients. Second, we distribute the three MobileNet family models to clients (**Model Zoo II**). In particular, we randomly assign MobileNetV1 to 16 clients, and MobileNetV2 and MobileNetV3 are randomly sent to the remaining 34 clients evenly. Finally, we use the four CNNs as the client models (**Model Zoo III**). Each CNN1, CNN2, and CNN3 model is randomly assigned to 12 clients. The remaining 14 clients will use CNN4 as the client model.

**Implementation Details**. We run all the experiments on NVIDIA A100 with CUDA version 12.0 on a Ubuntu 20.04.6 LTS server. All baselines and the proposed `pFedClub` are implemented in Pytorch 2.0.1. For the proposed `pFedClub` and baseline pFedHR, we set the number of clusters $K = 4$ following [23], and the local training epoch and the server finetuning epoch are equal to 10 and 3, respectively. The hyperparameter $\lambda$ in Eq. (5) is 0.2. The hyperparameter $\tau$ in Eq. (3) is 0.07. We use Adam as the optimizer. The learning rate of the local client learning and the server fine-tuning learning rate equal 0.001. We use **average client accuracy** with three runs as the evaluation metric.

### 4.2 Performance Comparison

**Heterogenous Model Aggregation**. Table 1 lists the experimental results regarding the **three-run accuracy** of the proposed `pFedClub` and baselines. Note that HeteroFL and FlexiFed belong to

---

[4] https://yann.lecun.com/exdb/mnist/
[5] http://ufldl.stanford.edu/housenumbers/
[6] https://www.cs.toronto.edu/~kriz/cifar.html

Table 1: Performance (%) comparison with baselines under the heterogeneous settings.

| Setting | Public Data | Method | MNIST IID | MNIST Non-IID | SVHN IID | SVHN Non-IID | CIFAR-10 IID | CIFAR-10 Non-IID |
|---|---|---|---|---|---|---|---|---|
| Model Zoo I | Labeled | FedMD | $91.12 \pm 2.44$ | $90.03 \pm 2.98$ | $76.22 \pm 3.01$ | $75.14 \pm 3.75$ | $66.38 \pm 3.96$ | $63.10 \pm 4.75$ |
| | | FedGH | $92.76 \pm 1.93$ | $91.27 \pm 2.21$ | $78.41 \pm 2.65$ | $75.06 \pm 2.87$ | $71.22 \pm 2.79$ | $67.37 \pm 3.06$ |
| | | pFedHR | $\mathbf{94.67} \pm 1.58$ | $92.88 \pm 1.10$ | $81.59 \pm 1.40$ | $80.88 \pm 1.92$ | $73.21 \pm 3.24$ | $69.88 \pm 3.45$ |
| | | pFedClub | $94.02 \pm 1.41$ | $\mathbf{93.20} \pm 0.85$ | $\mathbf{84.55} \pm 1.17$ | $\mathbf{82.65} \pm 1.56$ | $\mathbf{76.45} \pm 2.87$ | $\mathbf{73.62} \pm 3.01$ |
| | Unlabeled | FedKEMF | $91.47 \pm 1.87$ | $90.60 \pm 1.68$ | $77.56 \pm 2.47$ | $74.23 \pm 2.77$ | $68.77 \pm 2.54$ | $65.09 \pm 3.12$ |
| | | FCCL | $91.09 \pm 2.05$ | $90.21 \pm 2.44$ | $79.44 \pm 2.33$ | $75.28 \pm 2.60$ | $66.85 \pm 2.66$ | $64.76 \pm 2.98$ |
| | | pFedHR | $92.15 \pm 1.69$ | $91.00 \pm 1.73$ | $80.66 \pm 2.17$ | $78.93 \pm 2.55$ | $72.06 \pm 2.38$ | $68.54 \pm 2.47$ |
| | | pFedClub | $\mathbf{93.72} \pm 1.90$ | $\mathbf{92.77} \pm 1.51$ | $\mathbf{83.94} \pm 2.08$ | $\mathbf{81.76} \pm 2.32$ | $\mathbf{75.86} \pm 1.98$ | $\mathbf{72.87} \pm 2.04$ |
| Model Zoo II | Labeled | FedMD | $91.98 \pm 0.76$ | $92.01 \pm 1.05$ | $80.86 \pm 1.26$ | $77.53 \pm 1.53$ | $68.55 \pm 1.89$ | $63.74 \pm 2.25$ |
| | | FedGH | $92.13 \pm 1.32$ | $91.14 \pm 1.59$ | $78.15 \pm 1.50$ | $75.47 \pm 1.98$ | $71.29 \pm 1.77$ | $68.60 \pm 2.42$ |
| | | pFedHR | $93.51 \pm 1.36$ | $92.77 \pm 1.24$ | $82.33 \pm 1.86$ | $80.96 \pm 1.90$ | $73.60 \pm 2.38$ | $71.14 \pm 2.76$ |
| | | pFedClub | $\mathbf{93.62} \pm 1.07$ | $\mathbf{93.11} \pm 1.47$ | $\mathbf{84.25} \pm 2.04$ | $\mathbf{82.47} \pm 1.66$ | $\mathbf{76.02} \pm 1.53$ | $\mathbf{73.15} \pm 1.97$ |
| | Unlabeled | FedKEMF | $92.61 \pm 1.25$ | $91.33 \pm 1.70$ | $79.62 \pm 1.68$ | $77.54 \pm 2.04$ | $69.11 \pm 2.45$ | $66.07 \pm 2.88$ |
| | | FCCL | $92.77 \pm 1.77$ | $90.89 \pm 1.90$ | $81.88 \pm 1.79$ | $77.32 \pm 1.68$ | $68.02 \pm 2.06$ | $66.43 \pm 2.96$ |
| | | pFedHR | $93.21 \pm 1.45$ | $92.77 \pm 1.69$ | $81.56 \pm 1.50$ | $79.68 \pm 1.79$ | $71.88 \pm 1.75$ | $69.94 \pm 2.24$ |
| | | pFedClub | $\mathbf{93.86} \pm 1.34$ | $\mathbf{93.41} \pm 1.85$ | $\mathbf{83.89} \pm 1.22$ | $\mathbf{81.61} \pm 1.47$ | $\mathbf{75.52} \pm 1.39$ | $\mathbf{72.31} \pm 1.98$ |
| Model Zoo III | ✗ | HeteroFL | $92.48 \pm 1.14$ | $91.25 \pm 1.45$ | $80.57 \pm 1.37$ | $77.60 \pm 1.68$ | $71.08 \pm 1.57$ | $67.87 \pm 1.66$ |
| | | FlexiFed | $91.08 \pm 1.52$ | $90.10 \pm 1.66$ | $80.69 \pm 1.39$ | $75.30 \pm 1.62$ | $68.09 \pm 2.79$ | $67.15 \pm 2.88$ |
| | Labeled | FedMD | $92.16 \pm 1.32$ | $91.37 \pm 1.56$ | $80.22 \pm 1.59$ | $76.14 \pm 1.86$ | $67.14 \pm 1.67$ | $63.50 \pm 1.88$ |
| | | FedGH | $92.93 \pm 1.32$ | $91.44 \pm 1.08$ | $79.03 \pm 1.44$ | $75.28 \pm 1.75$ | $67.88 \pm 1.75$ | $70.77 \pm 1.93$ |
| | | pFedHR | $92.25 \pm 1.93$ | $91.07 \pm 1.64$ | $81.88 \pm 2.36$ | $79.25 \pm 1.71$ | $72.45 \pm 1.81$ | $69.08 \pm 1.95$ |
| | | pFedClub | $\mathbf{93.24} \pm 1.36$ | $\mathbf{92.66} \pm 0.98$ | $\mathbf{82.69} \pm 1.61$ | $\mathbf{81.21} \pm 1.68$ | $\mathbf{74.88} \pm 2.02$ | $\mathbf{71.94} \pm 1.82$ |
| | Unlabeled | FedKEMF | $92.78 \pm 0.75$ | $91.60 \pm 1.03$ | $78.88 \pm 1.68$ | $76.16 \pm 1.77$ | $68.04 \pm 2.16$ | $65.80 \pm 2.84$ |
| | | FCCL | $92.65 \pm 1.84$ | $91.07 \pm 1.92$ | $80.32 \pm 1.71$ | $76.02 \pm 1.82$ | $67.16 \pm 2.42$ | $66.73 \pm 2.68$ |
| | | pFedHR | $\mathbf{93.84} \pm 1.25$ | $93.46 \pm 1.50$ | $81.76 \pm 2.12$ | $78.40 \pm 2.50$ | $71.74 \pm 1.68$ | $68.23 \pm 1.79$ |
| | | pFedClub | $93.77 \pm 1.16$ | $\mathbf{93.52} \pm 1.37$ | $\mathbf{82.50} \pm 1.25$ | $\mathbf{81.03} \pm 1.49$ | $\mathbf{74.71} \pm 1.40$ | $\mathbf{71.68} \pm 1.59$ |

the submodel training technique and require that each client model must be a part of the global model. Thus, they are only tested with Model Zoo 3. These results show that our proposed approach outperforms all the baselines under most settings, especially the more complicated datasets SVHN and CIFAR-10. Compared with the most recent work pFedHR, our proposed work pFedClub shows superior performance over that on SVHN and CIFAR-10 datasets under both the IID and non-IID settings. For pFedClub, the use of the labeled public data is able to boost the performance compared with the setting using unlabeled public data, which aligns with the observations in [23]. In addition, from Model Zoo III to Model Zoo I, the performance of pFedClub on the more complicated datasets, SVHN and CIFAR-10, improves with the increase of diversity of the model zoos.

**Homogeneous Model Aggregation**. The clients are assigned the same model structures to verify the effectiveness of pFedClub under the homogenous setting. We test the performance with CNN2 and MobileNetV2 under the non-IID setting and compare it with the state-of-the-art homogenous federated learning work. The results are shown in Table 2. Note that pFedHR and pFedClub are under the setting where the public data is labeled. We observe that the results of all the approaches on the MNIST dataset are relatively high, even with a simple CNN2 model, as classification on the MNIST dataset is an easy task. Besides, pFedClub outperforms state-of-the-art baselines on SVHN and CIFAR-10 datasets using the CNN2 model or MobileNetV2 model. These results demonstrate that pFedClub is also effective for the homogeneous setting.

## 4.3 Ablation Study

We conduct the ablation study to validate the effectiveness of each designed module in our proposed approach with Model Zoo III under the non-IID setting. In particular, we use the following four baselines: (1) $\text{pFedClub}_{max}$: in the anchor block selection stage (Step 1 in Section 3.2.2), we naively select the most similar block calculated by Eq. (1) for the first block, instead of randomly selecting one. (2) $\text{pFedClub}_{min}$: different from $\text{pFedClub}_{max}$, we use the block with the smallest index number as the substitution. The substituted block is either itself or other models' first block. (3) $\text{pFedClub}_{noc}$: we conduct the block search without using the order con-

Table 2: Performance (%) comparison with baselines under the homogeneous setting.

| Model | Approach | MNIST | SVHN | CIFAR-10 |
|---|---|---|---|---|
| CNN2 | FedAvg | 90.52 | 62.49 | 58.01 |
| | FedProx | 90.87 | 63.77 | 59.65 |
| | Per-FedAvg | 91.04 | 63.59 | 59.81 |
| | PFedMe | 91.79 | 64.27 | 60.14 |
| | PFedBayes | 92.54 | 63.19 | 60.08 |
| | pFedHR | **92.62** | 64.59 | 61.79 |
| | pFedClub | 92.18 | **66.97** | **64.25** |
| MobileNetV2 | FedAvg | 92.07 | 79.42 | 62.13 |
| | FedProx | 92.85 | 80.33 | 63.60 |
| | Per-FedAvg | 92.62 | 82.45 | 70.88 |
| | PFedMe | 93.05 | 81.79 | 72.13 |
| | PFedBayes | **93.71** | 83.05 | 72.44 |
| | pFedHR | 93.15 | 83.88 | 73.65 |
| | pFedClub | 93.68 | **85.26** | **74.97** |

straint in Step 2; and (4) $\texttt{pFedClub}_{nbc}$: we do not conduct the block completion process (Step 3 in Section 3.2.2).

We report the results in Table 3 and provide the following observations: (1) Removal of any one module will cause the performance drop, thus demonstrating the individual contribution of each design in our proposed $\texttt{pFedClub}$. (2) When we use simple strategies (i.e., $\texttt{pFedClub}_{max}$ and $\texttt{pFedClub}_{min}$) in the anchor block selection, the drop in the performance is smaller than studies (i.e., $\texttt{pFedClub}_{noc}$ and $\texttt{pFedClub}_{nbc}$). It indicates that maintaining the block number and keeping the model completion matter more than the anchor block selection. Overall, each designed module has its own contribution, and the systematic combination of all the designed modules guarantees the effectiveness of our proposed $\texttt{pFedClub}$.

Table 3: Ablation study performance (%) comparison.

| Dataset | SVHN | | CIFAR-10 | |
|---|---|---|---|---|
| Method | IID | Non-IID | IID | Non-IID |
| $\texttt{pFedClub}_{max}$ | 78.69 | 73.50 | 70.52 | 66.89 |
| $\texttt{pFedClub}_{min}$ | 77.50 | 72.09 | 69.96 | 66.84 |
| $\texttt{pFedClub}_{noc}$ | 65.26 | 61.08 | 62.19 | 58.14 |
| $\texttt{pFedClub}_{nbc}$ | 63.01 | 59.47 | 61.38 | 56.02 |
| $\texttt{pFedClub}$ | **82.50** | **81.03** | **74.71** | **71.68** |

### 4.4 Controllability Analysis

**Experimental Setups.** The major advantage of the proposed $\texttt{pFedClub}$ is enabling the generation of controllable personalized candidate models. To clearly exhibit the insights, we use *Model Zoo I* as the heterogeneous model set, and each type of model is assigned to a corresponding client. Besides, each client will be mandatorily active during all the communication rounds. We use unlabeled public data for model training under the non-IID setting. To quantitatively evaluate the controllability of the personalized models generated by $\texttt{pFedClub}$, we propose to use the average of the model size change percentage over $T$ communication rounds as the metric for each client, which is defined as follows:

$$\phi = \frac{1}{T} \sum_{t=1}^{T} \frac{|\mathbf{w}*_m^t| - |\mathbf{w}_m|}{|\mathbf{w}_m|}, \tag{6}$$

where $|\mathbf{w}*_m^t|$ denotes the parameter size of the personalized teacher model at round $t$, and $|\mathbf{w}_m|$ denotes the model parameter size of the original client model.

**Model Comparison**. Since the proposed $\texttt{pFedClub}$ is a model reassembly-based framework, for a fair comparison, we choose to use pFedHR as the baseline. Besides, as mentioned in Section 3.2.2 Step 2, the proposed $\texttt{pFedClub}$ is flexible to incorporate other constraints. In this experiment, we take the model size into consideration and denote the model as $\texttt{pFedClub}^+$. The reason is that for real-world FL applications, such as training a model with smart devices, their computational capability is limited. Larger personalized models may make these devices stop working. To facilitate flexible management of the generated model's size in $\texttt{pFedClub}^+$, we introduce an additional parameter, $\eta > -1$, which provides flexible controllability to decide the generated model size following the constraint: $|\mathbf{w}*_m^t| \leq (1 + \eta)|\mathbf{w}_m^t|$. In this experiment, we set $\eta = 0.1$.

**Results**. Figure 3 illustrates the comparative performance with respect to accuracy and the model size controllability of pFedHR, $\texttt{pFedClub}$, and $\texttt{pFedClub}^+$ on the SVHN dataset under non-IID conditions. We observe that: **(1)** Both $\texttt{pFedClub}$ and $\texttt{pFedClub}^+$ show a superior performance over pFedHR shown in Figure 3(a). **(2)** As for the model size control, we can observe that $\texttt{pFedClub}$ and $\texttt{pFedClub}^+$ both have better effectiveness over pFedHR in Figure 3(b). For example, For example, for client

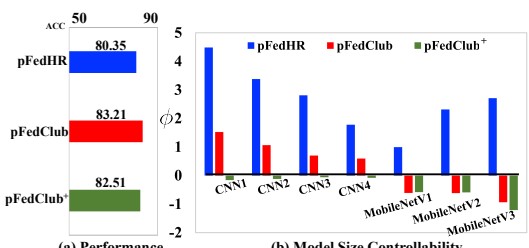

Figure 3: Controllability analysis.

1, the average received model parameter size is around 4.8 times as the original model for pFedHR and 1.5 times as the original model for $\texttt{pFedClub}$. For clients 5 - 7 using MobileNets, the average of the received personalized teacher model parameter size using $\texttt{pFedClub}$ is smaller than that of the original model size ($\phi < 0$). However, $\phi$ is still a large positive number for pFedHR, which means the clients still receive the personalized teacher models larger than their original ones. **(3)** When comparing $\texttt{pFedClub}$ with $\texttt{pFedClub}^+$, the latter shows a slight decrease in accuracy due to the rigorous model size constraint. Nonetheless, $\texttt{pFedClub}^+$ further refines the control over

the size of received personalized teacher models across all clients, ensuring they are not larger than than the original models ($\phi \leq 0$). This confirms the capability of both `pFedClub` and `pFedClub`$^{+}$ to effectively manage personalized model sizes, an essential feature for applications sensitive to computational resources. Additional details on model size comparisons between `pFedClub` and pFedHR across various communication rounds are provided in Appendix **C**.

### 4.5   Computational Cost Comparison

The proposed model can be treated as a fine-grained model reassembly technique, which uses controllable block-wise substitution to generate personalized candidates. In this experiment, we aim to compare the computational cost of the server between `pFedClub` and pFedHR using the computational time at each round on the SVHN dataset under the non-IID setting with Model Zoo III for 50 clients. We record the consumed computation time on the server side for each communication round. The results are shown in Figure 4. We can observe that the computation time at the server side of our approach `pFedClub` is generally shorter than that of the baseline pFedHR. Also, with the algorithm running with respect to the com-

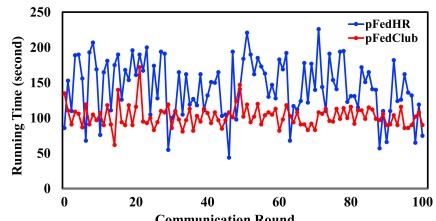

Figure 4: Server running time v.s. communication round.

munication round, our approach becomes more consistent and stable compared with the significant shift of pFedHR. These results confirm that `pFedClub` is an efficient approach for heterogeneous model aggregation compared with pFedHR.

### 4.6   System Running Time v.s. Accuracy

Except for controllability and computational costs, system running time is another key factor to evaluate the utility of the proposed `pFedClub`. Toward this end, we conduct an experiment to compare the consumed time to reach a fixed accuracy. The experimental setting is the same as the one that we described in Section 4.5. We take pFedHR as a baseline for comparison again. The results are shown in Figure 5. We can

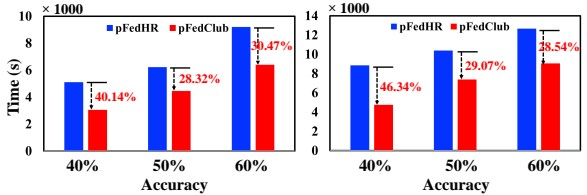

Figure 5: The consumed running time (in seconds) of models to achieve the target accuracy.

observe that the proposed approach `pFedClub` takes less time to achieve the target accuracy on the SVHN and CIFAR-10 datasets under the non-IID setting compared with pFedHR. These results demonstrate the effectiveness of the proposed `pFedClub` for the heterogeneous model aggregation in federated learning again.

### 4.7   Extra Experimental Results

To validate the **model scalability** of `pFedClub`, we conduct the experiments by considering different numbers and different active ratios of clients, and the results are shown in Table 4 in Appendix **D**. Besides, in our model design, there is a key parameter $K$ used in Section 3.2.1. We validate the sensitivity of the selection of $K$, and the results are listed in Table 5 in Appendix **E**.

## 5   Conclusion

This paper introduces `pFedClub` designed to revolutionize personalized federated learning. By leveraging a unique network block substitution method, `pFedClub` effectively creates tailored and functionally analogous personalized models for individual clients. Moreover, `pFedClub` is highly adaptable and can easily incorporate other types of constraints to achieve application-driven personalized model generation. Our experimental evaluations, conducted on three diverse datasets under both IID and non-IID settings, unequivocally validate the efficacy of `pFedClub` in the domain of heterogeneous model aggregation for federated learning. The results affirm the accuracy, efficiency, and flexibility of our proposed method, demonstrating its potential for real-world applications.

**Acknowledgements** This work is partially supported by the National Science Foundation under Grant No. 2348541 and 2238275.

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

# Appendix

## A. Baselines

In the heterogeneous experiments, we use the following approaches as baselines:
(1) Without using public datasets:

- HeteroFL [24]: Local clients' models are required to belong to the same model class and work together to produce one single global inference mode. Specifically, they revise the batch normalization, conduct a pre-activity scaling, and design a masked loss to solve this research problem.
- FlexiFed [25]: The common parts of the clients' models work together, and the different parts work separately to keep the system updated. They provide a basic-common strategy, cluster-common strategy, and max-common strategy to conduct the heterogeneous model aggregation under the setting.

(2) Using labeled public data:

- FedMD [16]: It uses transfer learning and knowledge distillation with the labeled public data on the server side. Specifically, each client needs to train the local model on both the public dataset and the private dataset. Then, clients upload the class scores on the public dataset to the server, and the server calculates the consensus to send it back for a local update.
- FedGH [19]: The clients have their own feature extractors and share the homogeneous global header. The clients train local models on their local data and upload the representation and label for each label back to the server for the global header update. Then, the clients replace their own headers with the global one for inference.
- pFedHR [23]: This approach utilizes the model disassembly techniques to decompose local models into layers. The server composes the layers back and tunes the models while stitching the layers using the public datasets.

(3) Using unlabeled public data:

- FCCL [17]: FCCL leverages the unlabeled public data and averages the logits from local clients. The approach utilizes a consensus logit to guide the local training.
- FedKEMF [21]: This approach aggregates knowledge from local models and distills it into global knowledge via knowledge distillation. It uses mutual learning to personalize the models on the server side with the unlabeled public data;
- pFedHR [23]: This approach is also able to use the unlabeled public data to tune the stitching layers of the candidate models.

The following baselines are used in the homogeneous experiments:

- FedAvg [1]: It is the vanilla version of federated learning, which averages the model parameters from the local clients;
- FedProx [27]: It adds the proximal term to the local model training based on FedAvg;
- Per-FedAvg [28]: The MAML framework is proposed based on meta learning;
- PFedMe [29]: It uses regularized loss and decouples the personalization problem into a bi-level optimization;
- PFedBayes [37]: It proposes an algorithm to take consideration of the global distribution while conducting local model training;
- pFedHR [23]: It generates personalized models by model decomposition and composition for local clients to guide local model training.

## B. CNN Strutures

In our experiments, we have 4 CNN models with different complexity. The details are shown as follows. In each convolutional NN sequential block, there is 1 convolutional layer, a max pooling layer, and a ReLU function.

**CNN1**: *Cov1:{Conv2d (kernel size = 5) → ReLU → MaxPool2D (kernel size = 2, stride = 2) } → Cov2:{Conv2d (kernel size = 5) → ReLU → MaxPool2D (kernel size = 2, stride = 2) } → FC1:{Linear → ReLU} → Dropout→ FC2:Linear.*

**CNN2**: *Cov1:{Conv2d (kernel size = 5) → ReLU → MaxPool2D (kernel size = 2, stride = 2) } → Cov2:{Conv2d (kernel size = 5) → ReLU → MaxPool2D (kernel size = 2, stride = 2) } → Cov3:{Conv2d (kernel size = 5) → ReLU} → FC1:{Linear → ReLU} → Dropout→ FC2:Linear.*

**CNN3**: *Cov1:{Conv2d (kernel size = 5) → ReLU → MaxPool2D (kernel size = 2, stride = 2) } → Cov2:{Conv2d (kernel size = 5) → ReLU → MaxPool2D (kernel size = 2, stride = 2) } → Cov3:{Conv2d (kernel size = 5) → ReLU} → Cov4:{Conv2d (kernel size = 5) → ReLU → MaxPool2D (kernel size = 2, stride = 2) } → Cov5:{Conv2d (kernel size = 5) → ReLU} → FC1:{Linear → ReLU → Dropout} → FC2:{Linear → ReLU}→ FC3:{Linear → ReLU}→FC4:Linear.*

**CNN4**: *Cov1:{Conv2d (kernel size = 5) → BatchNorm2d → ReLU} → Cov2:{Conv2d (kernel size = 3) → ReLU → MaxPool2D (kernel size = 2, stride = 2) } → Cov3:{Conv2d (kernel size = 3) → BatchNorm2d → ReLU} → Cov4:{Conv2d (kernel size = 5) → ReLU → MaxPool2D (kernel size = 2, stride = 2) → Dropout } → Cov5:{Conv2d (kernel size = 3) → BatchNorm2d → ReLU } → Cov6:{Conv2d (kernel size = 3) → ReLU → MaxPool2D (kernel size = 2, stride = 2) } → FC1:{Linear → ReLU → Dropout} → FC2:{Linear → ReLU}→ FC3:{Linear → ReLU}→FC4:Linear.*

## C. Generated Model Size Comparison

In Section 4.4, we have validated that the size of the generated personalized models by the proposed `pFedClub` is much smaller than that produced by pFedHR. Figure 3 shows the relatively average change in model size. In this experiment, we aim to show a detailed comparison of model size changes at each communication round for each type of heterogeneous model.

The results are shown in Figure 6. We can observe that our approach `pFedClub` conducts better control over the generated models compared with pFedHR in almost every communication round, especially for the client with the MobileNet models, where the size of our generalized model parameters is always smaller than the original one ($\phi < 0$).

## D. Model Scalability Analysis

Besides, model scalability is important in federated learning systems. To validate the scalability of `pFedClub`, we conduct the following experiments by considering different numbers and different active ratios of clients. The results are shown in Table 4.

Given the different settings of the different numbers of clients and active ratios, we can observe that the performance changes according to our expectations.

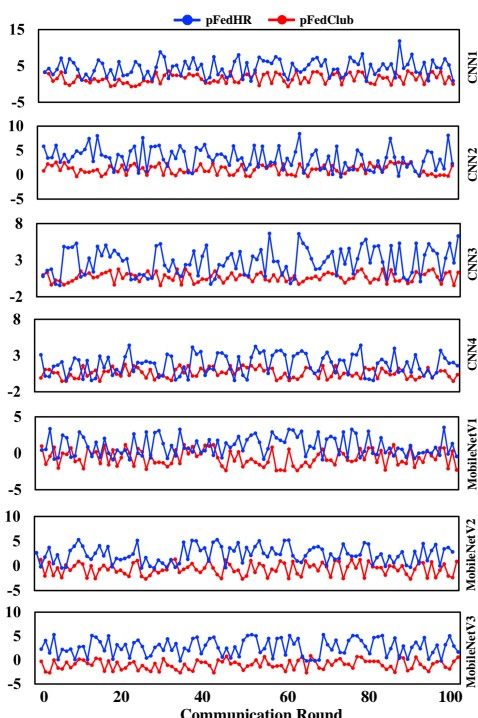

Figure 6: Model size controllability compared with pFedHR with respect to the communication round.

Moreover, given a fixed active ratio, the performance has a slight decrease with the increase in the client number. One possible reason is each client will have a smaller number of training data when the client number increases given a certain number of training data. The change in the local model performance will harm the performance of the whole system in a certain range. Furthermore, given a fixed number of clients, the increase in the active ratio will slightly boost the performance because it enables more clients to contribute to the update process with their local training at each communication round. The experiment results demonstrate the scalability of our proposed approach, considering the change in the client number and active ratio.

Table 4: Scalbility of `pFedClub` with different numbers of clients and different active ratios on the SVHN dataset.

| Data | IID | | | non-IID | | |
|---|---|---|---|---|---|---|
| Client | Active Ratio | | | Active Ratio | | |
| Number | 10% | 20% | 30% | 10% | 20% | 30% |
| 30 | 82.84 | 84.68 | 85.02 | 81.23 | 82.58 | 84.47 |
| 50 | 82.50 | 84.02 | 84.89 | 81.03 | 82.41 | 83.15 |
| 100 | 79.26 | 82.55 | 83.97 | 77.06 | 79.43 | 80.22 |

Table 5: Hyperparameter study of the number of clusters. The performance (%) of `pFedClub` with different values of $K$ on the SVHN and CIFAR-10 datasets.

| Public | Dataset | SVHN | | CIFAR-10 | |
|---|---|---|---|---|---|
| Dataset | Cluster | IID | Non-IID | IID | Non-IID |
| | 3 | 80.45 | 78.21 | 72.02 | 68.04 |
| Unlabeled | 4 | 82.50 | 81.03 | 74.71 | 71.68 |
| | 5 | 82.66 | 81.17 | 75.26 | 71.89 |
| | 3 | 80.76 | 79.88 | 73.07 | 68.95 |
| Labeled | 4 | 82.69 | 81.21 | 74.88 | 71.94 |
| | 5 | 82.77 | 81.35 | 75.89 | 72.38 |

## E. Hyperparameter Study

In our design, $K$ is the number of the groups based on the function-wise clustering with K-means. In this experiment, we aim to study how the hyperparameter $K$ affects the performance. We maintain the experimental setting in Sections 4.5 and 4.6. The results are shown in Table 5. We can observe that the performance of `pFedClub` will slightly increase with the increase of $K$. In the main experiments, we follow [23] and set $K = 4$. When $K = 5$, the performance can increase slightly. One possible reason is that a large $K$ is able to produce more specific function-based groups and further identify the function of the blocks more accurately. We can also observe that the performance is generally stable with the change of hyperparameter $K$ in a certain range.

## F. Limitations and Broader Impacts

This work focuses on the controllable personalized model generation for the heterogeneous federated learning setting. Although the proposed `pFedClub` outperforms baselines and is able to generate size-controllable client models, it still has several limitations. First, the proposed controllable model searching and reproduction (CMSR) algorithm is a heuristic algorithm that is designed based on intuitions. Thus, the results may be suboptimal. Second, in the system running time experiments, we have demonstrated that the proposed `pFedClub` can reduce the learning time compared with the state-of-the-art model pFedHR. However, compared with traditional averaging methods, the running time on the server is still much longer. We plan to design a more efficient algorithm for the server update. Finally, in the experiments, we only test the image classification task, which limits the efficacy test on other tasks. We will test more diverse tasks with the proposed `pFedClub` in the future.

This research significantly enhances federated learning by enabling efficient aggregation of heterogeneous models without compromising data privacy. This approach not only improves user experience across diverse sectors by providing tailored solutions but also supports sustainable computing practices of large-scale machine learning operations.

