# OpenReview forum: "pFedClub: Controllable Heterogeneous Model Aggregation for Personalized Federated Learning"
_NeurIPS.cc/2024/Conference — NeurIPS 2024 poster_

### Official Review · Reviewer_WRnm · 2024-06-29

**Soundness:** 4
**Presentation:** 3
**Contribution:** 3
**Rating:** 7
**Confidence:** 5

**Summary:**

The authors address a key issue in personalized federated learning, which enables clients with heterogeneous model structures to participate in federated learning with consideration of effectiveness and efficiency. This method is based on model assembly and reassembly, in which the blocks and layers can be treated as modules. After that, the server selects the personalized models and assigns them to the clients. The received models will be used as the teacher to guide the local update. The authors run extensive experiments to demonstrate the effectiveness of their algorithm.

**Strengths:**

1. This paper is well-organized and clearly motivated. Its logical structure and presentation aid comprehension, while the clear and accessible framework and figures enhance readability. Experiments, discussions, or analyses robustly support each claim.

2. The focus on controllability renders the algorithm more applicable in real-world scenarios, allowing for greater human involvement in the model generation process. The authors effectively demonstrate the utility of their design through experimental results.

3. The authors have performed extensive experiments, including principal studies on image datasets, ablation studies, hyperparameter evaluations, and thorough discussions. These efforts confirm the validity of the techniques and provide deep insights into the paper's contributions.

**Weaknesses:**

1. Based on the algorithm itself, it includes the reassembly, assembly, matching, and other operations. The reviewer may be concerned about the computational burden compared with the one without any controllability.

2. How to select the anchor block and why needs to be stated clearly.

3. According to the experiment results, the reviewer is wondering about how this approach can be used with the public data with/without labels and the possible reason why it is robust to the public data with or without labels.

**Questions:**

Please see the above weaknesses.

**Limitations:**

No negative social impact to the reviewer’s best knowledge. The study of K should be put in the main content as that is an important part of the algorithm.

---

> ### Author Rebuttal · Authors · 2024-08-05
>
> `>>> W1`
>
> Thank you for the reviewer's valuable feedback. We addressed the computational cost comparison in Section 4.5, using computation time as a metric against pFedHR. The results, presented in Figure 4, show that pFedClub generally requires less time than pFedHR and offers more consistent performance. Additionally, we detailed the system running time comparisons with the baseline in Appendix E. Our findings demonstrate that to achieve the target accuracy, our proposed approach is more time-efficient than pFedHR, further underscoring its efficiency.
>
> `>>> W2`
>
> Thank you for the question. The anchor block selection process is detailed from line 162 to line 169. For each block, we implement a random substitution, with the first substituted block designated as the anchor. This randomness enhances the diversity of candidate model generation. For a clearer explanation of this process, we have included a detailed algorithm in Appendix A, from lines 3 to 7.
>
> `>>> W3`
>
> Thank you for the reviewer's question. On the server side, public data is used solely to assist in (1) calculating the representations of the blocks for function identification and clustering, and (2) conducting stitching layer tuning. These operations utilize the data as foundational information without exploiting any specific attributes of the data itself. Consequently, our proposed approach is compatible with both labeled and unlabeled public data.

---

> > ### Comment · Reviewer_WRnm · 2024-08-13
> >
> > Thanks for your reply. I will keep my score positive as it is. Thank you.

---

### Official Review · Reviewer_PfmG · 2024-07-04

**Soundness:** 3
**Presentation:** 4
**Contribution:** 3
**Rating:** 7
**Confidence:** 5

**Summary:**

This paper presents a controllable model reassembly approach to enable heterogeneous model cooperation in federated learning. The designed CMSR algorithm provides the control of the space to save the computational cost.  Furthermore, the approach also achieves model personalization for each local client. They test the proposed approach on benchmark datasets and compare with other baselines under different settings.

**Strengths:**

1, This paper targets one of the challenges in federated learning, which is the model heterogeneity. To the best knowledge, most existing related works are based on knowledge distillation.  This work presents a controllable approach to conduct block assembly and reassembly from local models to achieve heterogeneous model cooperation and model personalization. The idea itself is interesting and practical.
2, They take efficiency, generalization, and personalization into consideration. They provide comprehensive analysis and provide detailed discussion under various settings, which support their statement soundly.
3, Their presentation and logic are both easy to follow and understand. The framework, experiment results, and discussion are clearly presented.

**Weaknesses:**

Weakness
1, In their approach, the authors employ K-means clustering. The reviewer is curious about how the value of K is selected and how this selection influences the results.

2, One of the main contributions compared to pFedHR is the enhanced controllability. I am interested in understanding the nature of this controllability, specifically the extent to which the generated models can be controlled.

3, The paper focuses solely on image classification. Adhering to the review guidelines, the reviewer is not requesting additional experiments, but the reviewer is interested in exploring whether the existing methodology could be applicable to other tasks.

**Questions:**

1, Please address the concerns in the weakness part.
2, After the blocks are stitched, the parameters of the blocks and/or the stitching layer would be trained? If trained, how are they trained? If not, how do you deal with the parameters of the stitching layers?

**Limitations:**

There are no potential negative societal impacts of the work. There are two limitations as follows:
(1)	This approach still raises extra computational cost at the server side.
(2)	That would be great if this approach can be extended to other tasks and other domains.

---

> ### Author Rebuttal · Authors · 2024-08-05
>
> `>>> W1`
>
> Thank you for the reviewer’s question. In the main paper, we set K to 4, following the methodology outlined in [1]. To further explore how the value of K affects our results, we conducted a hyperparameter study on K. The results of this study are presented in Table 5, with detailed analysis provided in Appendix G due to page constraints. We observed that performance slightly improves with a larger value of K within a certain range. This improvement may be due to the fact that a larger K can differentiate functions more specifically.
>
> [1] Wang et al. "Towards personalized federated learning via heterogeneous model reassembly." Advances in Neural Information Processing Systems 36 (2024).
>
> `>>> W2`
>
> Thanks for the reviewer’s question. Our controllability is demonstrated in sec 3.2.3 controllable block-wise substitution. Furthermore, we provided the controllability analysis in section 4.4. We define a hyperparameter $\phi$ to quantify the controllability of the personalized models generated by pFedClub. Also, we add the flexible controllability parameter $\eta$ to decide the generated model size. The results are shown in Figure 3. Compared with pFedHR (shown in blue), the generated models by the pFedClub (shown in red) and pFedClub+(shown in green) are significantly smaller than the ones generated by pFedHR.
>
> `>>> W3`
>
> Thank you for the reviewer's valuable comments. As we mentioned in the limitations section, our experimental results are currently limited to image classification tasks. However, given the generalized design and intuitive nature of our approach, we believe it can be adapted to other tasks, provided that the models can be assembled and reassembled. We are committed to exploring these possibilities in future research directions.

---

> > ### Comment · Reviewer_PfmG · 2024-08-13
> >
> > Thanks for the reply. After reading the answers,  they have addressed my concerns. I would like to increase my score appropriately.

---

> > > ### Author Response · Authors · 2024-08-13
> > >
> > > We sincerely appreciate your time to check the rebuttal and your recognition of our work.

---

### Official Review · Reviewer_AXhq · 2024-07-11

**Soundness:** 2
**Presentation:** 2
**Contribution:** 2
**Rating:** 5
**Confidence:** 4

**Summary:**

The paper proposes a `pFedClub` method for personalized federated learning that enables controllable heterogeneous model aggregation, addressing limitations of existing approaches such as lack of personalization, privacy concerns, and uncontrolled model size growth.

Extensive experiments conducted on three benchmark datasets using various CNN-based model structures validate the effectiveness of the proposed method under both IID and non-IID settings.

**Strengths:**

- They conduct extensive experiment including the discussion about the hyparameter $K$ to validate the controllability of the proposed method and computational efficiency on the server.

**Weaknesses:**

1. The writing and structure of the paper need improvement, particularly in the "Order-constrained Block Search" paragraph. The concept of order is unclear, especially the meaning of $q < u$ in line 177. It's not evident whether this refers to a similarity score or another metric. The author should provide a clearer explanation of this constraint.
2. In equation (1) on line 141, the meaning of 'CKA' is not defined. The authors should explain what CKA stands for and how it's calculated. Additionally, it's unclear whether this computation occurs on the server. If clients must transmit input $x_{m,i}^t$ and output to the server, this raises privacy concerns that should be addressed.
3. The paper doesn't specify whether the features $x_{m,i}^t$ and $x_{n,j}^t$ in equation (1) have the same dimensions. This should be clarified to ensure a proper understanding of the similarity calculation.
4. The sampling process for the Anchor Block selection is ambiguous. The probability distribution over all models for this selection is not clearly defined.

Overall, the authors should formulate the proposed method more rigorously, using well-defined notations and providing clear explanations for each step of the algorithm. This would significantly improve the paper's readability and reproducibility.

**Questions:**

1. Table 5 indicates that the model achieves the best performance in both IID and non-IID settings when K equals the number of activated clients. However, this raises a question about the necessity of the K-means method in this scenario. When K equals the number of activated clients, the input data naturally satisfies the minimization objective of the K-means algorithm, rendering the clustering step redundant. It would be valuable to explore how K affects the results when the number of activated clients increases, for example, to 10 or 20. This analysis would provide deeper insights into the scalability and robustness of the proposed method.

2. While the current experiments focus on CNN-based structures, it would be beneficial to validate the proposed method on other neural network architectures. Specifically, evaluating pFedClub on Transformer-based structures would demonstrate its versatility and applicability across different model types. This expansion of the experimental scope would strengthen the paper's contributions and broaden its potential impact in the field of federated learning.

3. The supplementary material should include a comprehensive set of implementation details for all methods used in the comparisons. This should encompass not only the proposed pFedClub method but also such as the hyparameter for baseline approaches.

---

> ### Author Rebuttal · Authors · 2024-08-05
>
> `>>> W1`
>
> Thank you for pointing this out. We follow existing work [1] to maintain the natural order of the blocks, as we aim for the generated candidate models to be similar to handcrafted network structures. Here, the natural order index is defined as the position of each block. For example, CNN1 listed in Appendix Section C consists of four layers: Conv1, Conv2, FC1, and FC2, with corresponding block order indexes of 1, 2, 3, and 4. In the generated model, we do not expect the FC1 layer to appear before the two convolutional layers, Conv1 and Conv2. Therefore, we constrain the order of blocks when generating candidate models.
>
> We also have an example in Figure 2. In step 2: Order-constrained Block Search, if the first selection for the target network (in red) is the second block (in green), the next selected block for block 2 of the target network must have an index larger than 2. In this example, we select two “3” blocks as candidates for the second block replacement. Similarly, the third block replacement will use indexes “4” and “5”. We will add these details in the final version of our paper.
>
> [1] Yang et al. Deep model reassembly. NeurIPS, 2022.
>
> `>>> W2`
>
> Thanks for the reviewer’s suggestion. CKA represents the centered kernel alignment, which is a method generally used to calculate the similarity of the neural network representations [2]. Specifically, given two kernels of feature representations K and L, we first calculate the Hilbert-Schmidt Independence Criterion (HSIC) by HSIC(K,L) = $1/(n-1)^2$ tr(KHLH), where n is the dimension of the representation of the features, and H is the centering matrix. CKA(K,L) = HSIC(K,L)/$\sqrt{(HSIC(K,K)HSIC(L,L))}$. Due to the page limit, we did not add those specific details in the original paper. We will add them in the final version.
>
> As we stated in Sec. 3.2, all the operations including CKA occur on the server. The clients do not need to transmit x and output to the server. Instead, clients only transmit their model parameters to the server, following the conventional federated learning approach. Upon receiving these parameters, the server uses public data as input to the initial block of models, with the output of each block serving as input for the subsequent block. We have detailed the experimental settings with public data, demonstrating that our method performs effectively with both labeled and unlabeled public data, outperforming other baselines.
>
> [2] Kornblith et al. "Similarity of neural network representations revisited." ICML, 2019.
>
> `>>> W3`
>
> Thanks for the reviewer’s valuable question. To ensure the generalization of our approach, the dimensions of the two features do not have to be the same, and CKA is able to handle this situation based on what we have addressed in W2 [2].
>
> `>>> W4`
>
> Thanks for the reviewer’s suggestion. The anchor block selection process considers the groups G. For model $w_m^t$, assuming its the first block $B_{m,1}^{t}$ belongs to the group $G_k^t$, we randomly select one block from $G_{k}^{t}$ as the substitution. Any block from the group $G_k^t$ has the equal probability $1/G_k$ to be selected (Line 169), where $G_k$ is the number of the blocks belonging to $G_k^t$.  With such a strategy, compared with the strategy using the two blocks with the largest similarity score, it increases the diversity of the generalized candidates. At the same time, this random strategy is effective, generalized to all models, and simple to implement.  As the reviewer suggested, we will add more descriptions of the anchor block selection process in the final version.
>
> We do appreciate the reviewer’s suggestion. We introduced the proposed algorithm CMSR in the appendix A. As the reviewer suggested, we will further enhance the readability and reproducibility in the final version.
>
> `>>> Q1`
>
> The clustering processing is based on the functionality in Line 139 with no direct relation with the number of clients based on our design. With an increasing cluster number in some range, it indicates that we will get a more specific approach to differentiate the functions of each block, which may have some boost on the performance, as shown in Table 5.
>
>  We totally agree with the reviewer about the exploration of the activated client number. We have conducted the experiments and shown the results in Table 4. In Table 4, we maintain the same value of K = 4 as the main experiment. Then we study a different total number of clients and different active ratios under the IID and non-IID settings. Furthermore, we provided the experiment observations and respective model scalability analysis in Appendix F. Combining the results in Table 4 and Table 5 and its analysis in Appendix G and F, it demonstrates the scalability and robustness of our proposed method.
>
> `>>> Q2`
>
> Thanks for the valuable question. In our limitation part, we admitted that our existing approach has not been tested on other tasks, such as those with the transformer-based structures. We sincerely appreciate the constructive suggestions from the reviewer. We will add more discussion in our limitation part and expand our work following this direction in the future.
>
> `>>> Q3`
>
> We appreciate the reviewer's suggestion. Our experiment setup, detailed in Section 4.1, includes descriptions of the datasets, baselines, client model deployment, and implementation specifics. We have ensured that common parameters are consistently maintained across different baselines while retaining the unique parameters of each baseline at their default values. As suggested by the reviewer, we will include a more comprehensive set of implementation details in the final version of our paper.
>
> We are sincerely grateful for the reviewer's constructive questions and suggestions. We hope our responses have adequately addressed your concerns. Thank you once again for your valuable feedback, which has significantly contributed to enhancing the quality of our paper.

---

> > ### Author Response · Authors · 2024-08-11
> >
> > Dear Reviewer AXhq,
> >
> > We greatly appreciate your constructive comments, which have significantly clarified our paper. As the discussion phase nears its end, could you kindly confirm whether our responses have satisfactorily addressed your concerns?
> >
> > Thank you,
> >
> > Authors of Paper 3650

---

> > ### Comment · Reviewer_AXhq · 2024-08-13
> >
> > Thank you for your response. I am inclined to raise my score to 5. However, I recommend that the authors include more details and explanations to improve the readability of the paper.
> >
> > The approach to aggregating heterogeneous models is particularly interesting. I encourage the authors to validate the proposed method on a broader range of neural network architectures.

---

> > > ### Author Response · Authors · 2024-08-13
> > >
> > > Dear reviewer,
> > >
> > > Thanks for your time to consider our reply. We appreciate your valuable comments. We will add more details and explanations to improve the readability of our paper in the camera-ready version. Also, we will extend our proposed approach to a broader range of neural network architectures as the reviewer suggested.
> > >
> > > Thank you.

---

### Official Review · Reviewer_5tH7 · 2024-07-14

**Soundness:** 3
**Presentation:** 3
**Contribution:** 3
**Rating:** 8
**Confidence:** 4

**Summary:**

This paper addresses heterogeneous model aggregation in federated learning. To this end, the authors introduce pFedClub, which aims to generate personalized models for federated clients while ensuring that the models remain within size constraints. Specifically, pFedClub consists of three main steps: first, it decomposes models into multiple blocks and clusters them using the K-means algorithm; second, it replaces original blocks with others from the same clusters to create a set of candidate models; third, it selects the optimal personalized model for each client using a public dataset and an initial model transferred to the server. Extensive experiments illustrate its significant improvement over existing methods in this field.

**Strengths:**

1. The work is well motivated and explores an interesting problem in federated learning.
2. The presentation of this paper is clear, and the authors comprehensively and intuitively describe the proposed pFedClub.
3. The paper conducts sufficient experiments and compares the proposed method with previous works. The numerical results demonstrate the superiority of pFedClub.

**Weaknesses:**

1. The proposed work requires a public dataset, which is unsuitable in federated learning due to the privacy concerns. Is this work applicable to a public dataset different from the training data distribution? For example, the clients collaboratively train a model for CIFAR-10, while the server holds a public dataset from ImageNet.
2. Although the proposed work achieves remarkable under convolutional neural networks, it is unclear how pFedClub performs under transformers. Is the proposed work suitable for a setting where clients hold three different sizes of LLM, i.e., LLaMA-7B, LLaMA-13B, and LLaMA-70B?

**Questions:**

See **Weaknesses**

**Limitations:**

See **Weaknesses**

---

> ### Author Rebuttal · Authors · 2024-08-05
>
> `>>> W1`
>
> Thank you for your comments. We would like to emphasize that our research question addresses a significantly challenging problem, where each client maintains a unique model. Aggregating these heterogeneous models on the server is particularly difficult without the use of public data. Additionally, we adhere to established work that serves as baselines in our experiments, which also incorporate public data during model training. While we acknowledge that differences in data distributions between public and private data can affect performance, our model design mitigates this issue, similar to the approach taken by pFedHR.
>
> Following the reviewer’s suggestion, we conducted an additional experiment where clients train models using CIFAR-10, and the server utilizes a public dataset from ImageNet. All other hyperparameters were kept consistent with those listed in Table 1 for Model Zoo 1 under the non-IID setting. The results of this experiment are as follows:
>
> | Public Data | Labeled | Unlabeled |
> |-------------|---------|-----------|
> | CIFAR-10    | 73.62   | 72.87     |
> | ImageNet    | 71.50   | 69.33     |
>
> We observe that using ImageNet as the public dataset does decrease performance compared to using CIFAR-10 directly. However, the performance drop is limited, and it still outperforms the best baseline, pFedHR (Labeled: 69.88 and Unlabeled: 68.54), as shown in Table 1.
>
> `>>> W2`
>
> Thank you for your feedback. In our current work, we focused on reassembling convolutional neural networks in our experiments, as they consist of distinguishable layers. However, our model design is flexible and can also be applied to transformer-based models, as demonstrated in [1], which segments transformer models into functional blocks and reassembles them into candidates. We did not test the transformer or more advanced models mentioned by Reviewer 5tH7 due to the following reasons: This work focuses on controllable personalized model generation, specifically designed for resource-constrained clients. Training transformer-based models from scratch typically requires a large amount of training data, which may be impractical for clients using mobile devices due to computational costs. Therefore, we did not include experiments with transformers in this study. We can include additional results involving transformer models in the final version of our paper if required.
>
> [1] Yang, Xingyi, Daquan Zhou, Songhua Liu, Jingwen Ye, and Xinchao Wang. "Deep model reassembly." Advances in neural information processing systems 35 (2022): 25739-25753.

---

> > ### Comment · Reviewer_5tH7 · 2024-08-08
> >
> > Thanks for your response. I find the rebuttal addresses my concerns. I believe this is a good paper and benefits ML community. Therefore, I decide to increase my score from 6 to 8.

---

> > > ### Author Response · Authors · 2024-08-08
> > >
> > > Thank you for your feedback. We are delighted to have addressed your concerns and appreciate the improved score.

---

### Decision · Program_Chairs · 2024-09-25

**Decision:**

Accept (poster)

**Comment:**

In this paper, the authors present a new method for heterogeneous model aggregation in personalized federated learning. Now, the paper has received four reviews, and the reviewers all think that the paper has studied an important problem with strong results through extensive experiments.

While the paper shows some promise, there are some weaknesses that have been identified in the reviews.
(1) More experimental evaluations are suggested, eg, under public datasets.
(2) Some technical details require more clarifications.
(3) The presentation requires more improvements.

During the rebuttal, the authors provided detailed responses to each reviewer's comments, and the reviewers were generally convinced by the authors' clarifications. During the discussion period, each review showed positive score about this submission, therefore, we recommend to accept this paper in NeurIPS 2024.